# 3D Scene De-occlusion in Neural Radiance Fields: A Framework for Obstacle Removal and Realistic Inpainting

## ABSTRACT

Neural Radiance Fields (NeRFs) demonstrate high efficiency in generating photo-realistic novel view. Recent studies introduce the trials on the 3D inpainting by NeRF. However, the performance of these works have been validated for data collected in a narrow range of multi-view, while degrade for the wide range of multi-view. To address this problem, we propose a novel NeRF framework to remove the obstacle and reproduce occluded areas in high quality for both wide and narrow range of multi-view. In this framework, we design a region coding network to carry out object segmentation. With the depth information, the segmentation component transfers a single obstacle mask to other views in high accuracy. By referring to the segmentation results, we introduce an innovative view selection mechanism to reconstruct the occluded area using supplementary information from multi-view and 2D inpainting. We also contribute to the evaluation of 3D scene de-occlusion by introducing a dataset including views captured in wide range and in pair with and without the obstacle object for comparison. We evaluate our framework in both narrow and wide range datasets by quantitative measurement and visually qualitative comparison, which confirm the competitive and superior performance of our framework.

## CCS CONCEPTS

• **Computing methodologies** → Image-based rendering; *Computer vision tasks*; **Rendering**.

## KEYWORDS

Neural Radiance Fields (NeRFs), De-occlusion, 3D Object Segmentation

**ACM Reference Format:**
Anonymous Author(s). 2024. 3D Scene De-occlusion in Neural Radiance Fields: A Framework for Obstacle Removal and Realistic Inpainting. In *Proceedings of the 32nd ACM International Conference on Multimedia (MM'24), October 28-November 1, 2024, Melbourne, Australia.* ACM, New York, NY, USA, 9 pages. https://doi.org/10.1145/nnnnnnn.nnnnnnn

## 1 INTRODUCTION

In recent years, Neural Radiance Fields (NeRF) [21] have attracted widespread attention for the exceptional capabilities in novel view

synthesis. Subsequent studies have been contributed to the progresses of NeRF to improve the training speed [6, 11, 16, 28, 33], the rendered image quality [3, 34, 38], the support for dynamic scene modeling [30, 31], the interactive 3D scene editing [26, 27], the scene style conversion [5, 13], etc. One notable 3D editing application is de-occlusion that aims at removing unwanted objects and recovering occluded areas realistically. This task faces multiple challenges, such as precisely annotating occluded areas across views, manipulating the removal of obstacle in scenes represented by NeRF's implicit model, preserving the multi-view consistency and the realistic inpainting after de-occlusion.

Existing studies have made attempts for these issues, while more efforts for progress are still desired. In terms of annotating multi-view occluded areas, Cheng *et al.* [4] models the space-time correspondence in video object segmentation. The results for the remaining frames can be inferred from the segmented one of a given frame. However, this method does not guarantee the 3D view consistency. Zhi *et al.* [41] mark objects through scene semantic information, while this method is sensitive to the number of semantic frames. The performance degrades with fewer labels due to uncovered or occluded regions. In terms of NeRF-based inpainting tasks, Object-NeRF [35] attempts to achieve the restoration by decomposing the scene into background and foreground, whereas this method lacks effective supervision of occlusions and cannot guarantee reasonable removal of obstacles. CLIP-NeRF [29] manipulates the scene by using a pre-trained natural language model. The effect of removing occlusion by CLIP-NeRF is limited by the accuracy of the object detector. Recently, the studies [22, 24, 32] attempt to involve 2D inpainting techniques. Weder *et al.* [32] introduce a framework that requires users to provide object masks for all views. NeRF-In [24] overlooks the scene consistency after de-occlusion. Although SPIn-NeRF [22] takes into account view consistency, it is still easy to ignore details behind the obstacle, especially for scenes shot from a wide range of multi-view.

In this paper, we propose a new pipeline to segment the obstacle in multi-view images and achieve more realistic inpainting of occluded areas. Specifically, derived from a single mask of the target object, we introduce an object segmentation component to obtain the corresponding masks in other views. This segmentation component adopts a region coding network to determine whether a sampling point locates in the occluded area or not. Next, the inpainting task is completed in a designed 3D inpainting component where we propose a training view selection mechanism to select reasonable content for reference to recover the occluded area. The selection mechanism involves the regional average gradient to formulate the reconstruction loss function for evaluating the fidelity of supplementary information from a pretrained 2D inpainter [25] and other views. This mechanism takes advantage of appearance consistency across views and effectively reduce the disturbing of 2D inpainter for 3D editing in scenes supported by wide range of

multi-view. Contributed by the region coding network and training view selection mechanism, the proposed de-occlusion framework can identify the target object precisely in different views and accomplish the inpainting realistically.

The proposed method has been evaluated in different datasets to demonstrate its effectiveness in the obstacle localization and the inpainting of the occluded area through qualitative and quantitative assessments. The contributions of the paper are summarized as follows:

- We propose a new framework for obstacle removal in NeRF-modeled scene, requiring a single view mask to achieve accurate inpainting results;
- We introduce a new region encoding network to support the multi-view object segmentation following the 3D consistent;
- We introduce a training view selection mechanism to keep a high fidelity in inpainting of occluded area;
- We establish a new 3D de-occlusion dataset , which includes scene images with and without the obstacle. It supports a wide range of multi-view to capture scene information naturally presented in 3D space.

## 2 RELATED WORK

### 2.1 Image inpainting

2D image inpainting aims to produce visually plausible content for the missing regions of incomplete images [40]. Patch-based methods [8, 9] recover pixels from neighbor visible information. In order to fill in large regions, structural information and attention modules are commonly used in a progressive mode [23, 36]. As an ill-posed problem, the adversarial training [18, 42] and diffusion model [19] have been applied to the inpainting task. Recently, a video inpainting method [17] uses the temporal focal transformer to model long-range dependencies on both spatial and temporal dimensions. However, for the 2D restoration task, a number of visually plausible results can often be produced by differently designed constraints. In contrast, the inpainting in 3D space needs to ensure the muti-view consistency that verifies the fidelity of each view. In this paper, the proposed method recover the occluded areas in each view and keep high accuracy assisted by information from other perspectives. The advantage of the proposed 3D inpainting is more prominent for the scene captured in the wide range of the multi-view.

### 2.2 NeRF Manipulation

Recent years, much research has been devoted to improving the modeling efficiency of NeRF [3, 26, 27, 34, 38] and extending its applications [1, 2, 7, 14, 15, 37, 43]. Although these advancements have significantly enhanced the practicality of NeRF, the removal of obstacles in NeRF-modeled scenes remains an area not yet fully explored. Existing studies have attempted to divide the scene into static and transient objects through the network, thereby achieving the removal of transient or dynamic objects [20]. However, this method lacks the interactive control by the user during the removal. Object-NeRF [35] provides an editable scene rendering method, while it does not performs well in cluttered scenes. Later, NeRF-In

[24] and SPIn-NeRF [22] use 2D inpainting results as prior information to recover 3D scene views. NeRF-In [24] directly uses the 2D inpainting results as color priors to fill the occluded areas in some views, while does not address inconsistency problem. SPIn-NeRF [22] involves a perceptual loss function and tries to maintains the view-consistency. However, it does not give a proper measure for the use of the 2D inpainter. In comparison, our approach introduces a training view selection mechanism based on regional average gradients. This mechanism aims to avoid the degradation by the unreasonable use of 2D inpainting.

## 3 PROPOSED METHOD

Given a set of multi-view images $I = \{I_s | s = 1...K\}$ and their corresponding camera positions, our method first utilizes the NeRF [21] model to obtain depth information, which provides necessary scene modeling constraints for the following object segmentation component. Next, the proposed method requires the user to select one view $I_o$ and draw the object mask $M_o$ on $I_o$ to annotate the occluded area. Subsequently, we introduce an object segmentation network. Under the constraint of depth information, this network can automatically infer masks $M = \{M_s | s = 1...K\}$ for other views (Sec 3.2). After that, we design a 3D inpainting component that uses a 2D inpainter to initially recover occluded areas, and then modify the recovered areas in consideration of the correlation of the multi-view to improve the fidelity of the de-occluded results (Sec 3.3). Figure 1 presents the overview of the proposed framework.

### 3.1 NeRF Component

Neural Radiance Fields (NeRFs)[21] achieve high-quality novel view rendering results by continuously modeling scenes with an MLP network. In the NeRF model, the MLP network takes the 3D location $x$ and 2D viewing direction $d$ as inputs and produces the emitted color $c$ and volume density $\sigma$. Then, through volumetric rendering, the estimated color $\hat{C}(r)$ of a pixel is calculated by accumulating the emitted color of $N$ random quadrature points along the ray $r$:

$$\hat{C}(r) = \sum_{i=1}^{N}(T_i(1 - \exp(-\sigma_i\delta_i))c_i), \text{where } T_i = \exp(-\sum_{j=1}^{i-1}(\sigma_j\delta_j))$$
(1)

where $T_i$ is the transmittance, $\delta_i = t_{i+1} - t_i$ is the distance between two adjacent points. Then, a loss function is defined by comparing the rendered color $\hat{C}(r)$ and the ground truth $C(r)$: $\mathcal{L} = \sum_{r \in R}[||\hat{C}_c(r) - C(r)||_2^2 + ||\hat{C}_f(r) - C(r)||_2^2]$, where $\hat{C}_c(r)$ represents the predicted coarse volume, and $\hat{C}_f(r)$ represent the predicted fine volume. This loss function guides the optimization process of the implicit scene representation. With the trained NeRF model, we can obtain multi-view depth information $D = \{D_s | s = 1...K\}$, which provides important depth prior for the next object segmentation component.

### 3.2 Object Segmentation Component

In order to obtain accurate multi-view masks from a single one annotated by users, we design an object segmentation component that obtains the semantic information of whether sampling points belong to occluded areas under the guidance of the given mask,

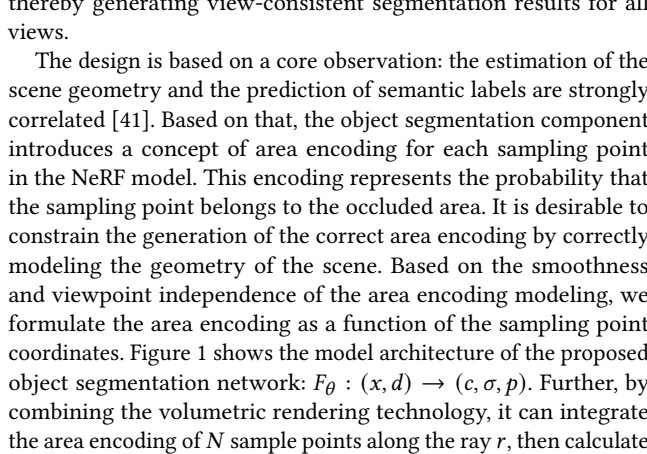

**Figure 1: An overview of the proposed framework. We input multi-view images with camera poses into the NeRF network, obtaining depth information of the input views. Next, a user marks the obstacle object on one view. Then, guided by the single mask, masks for all views are generated through a NeRF-based object segmentation component. Finally, the 3D inpainting component is used to achieve precise de-occlusion and ensure realistic inpainting results of occluded areas.**

thereby generating view-consistent segmentation results for all views.

The design is based on a core observation: the estimation of the scene geometry and the prediction of semantic labels are strongly correlated [41]. Based on that, the object segmentation component introduces a concept of area encoding for each sampling point in the NeRF model. This encoding represents the probability that the sampling point belongs to the occluded area. It is desirable to constrain the generation of the correct area encoding by correctly modeling the geometry of the scene. Based on the smoothness and viewpoint independence of the area encoding modeling, we formulate the area encoding as a function of the sampling point coordinates. Figure 1 shows the model architecture of the proposed object segmentation network: $F_\theta : (x, d) \rightarrow (c, \sigma, p)$. Further, by combining the volumetric rendering technology, it can integrate the area encoding of $N$ sample points along the ray $r$, then calculate the corresponding predicted mask value $\hat{P}(r)$ for each pixel:

$$\hat{P}(r) = \sum_{i=1}^{N} (T_i(1 - \exp(-\sigma_i \delta_i))p_i), \text{where } T_i = \exp\left(-\sum_{j=1}^{i-1} (\sigma_j \delta_j)\right) \tag{2}$$

During the optimization process, we employ two loss items: the color loss and the area encoding loss. The color loss ensures high consistency between the predicted color $\hat{C}(r)$ and the ground truth $C(r)$ for un-occluded areas, to confirm the correct modeling of scene geometry and appearance. On the other hand, at view $o$, the area encoding loss works by comparing the difference between the predicted mask value $\hat{P}(r)$ and the actual mask value $P(r)$ in image $I_o$. The two loss items are formulated bellow,

$$\mathcal{L}_{rgb} = \sum_{r \in R \wedge \hat{P}(r)=0} ||\hat{C}(r) - C(r)||_2^2 \tag{3}$$

$$\mathcal{L}_{area-encoding} = \sum_{r \in R_o} ||\hat{P}(r) - P(r)||_2^2 \tag{4}$$

where $R$ is the set of rays in the input views, $R_o$ is the set of rays in view $o$, $\hat{P}(r) = 0$ indicates that ray $r$ is in the non-occluded area of the rendered view. Optimizing the NeRF model under the joint constraints of these two losses not only models the geometry and appearance of the scene but also captures the semantic information of the scene. However, in terms of the occluded area, the geometric modeling process is supervised only by a single mask image, making it difficult for the model to capture the complex geometric features. Moreover, due to NeRF's entangled scene representation method, geometric modeling errors in the occluded area can affect the geometric and appearance modeling of non-occluded areas, thereby breaking NeRF's multi-view consistency. This results in inaccurate multi-view mask prediction results. To address this problem, we further add a depth loss. Utilizing depth information $D$ obtained from the NeRF Component to guide the geometric modeling of the occluded area. Specifically, we construct a depth loss by comparing the rendered depth $\hat{D}(r)$ of ray $r$ with the ground truth depth value $D(r)$:

$$\mathcal{L}_{depth} = \sum_{r \in R} ||\hat{D}(r) - D(r)||_2^2 \tag{5}$$

In summary, the object segmentation network is trained using the following loss: $\mathcal{L} = \lambda_1 \mathcal{L}_{region-encoding} + \lambda_2 \mathcal{L}_{rgb} + \lambda_3 \mathcal{L}_{depth}$, where $\lambda_1$, $\lambda_2$, and $\lambda_3$ are weights. The joint optimization strategy of the three losses ensures the multi-view consistency and smoothness of NeRF, thereby ensuring that accurate multi-view segmentation results obtained from a given single mask.

### 3.3 3D inpainting Component

Our method uses the 2D image inpainter, LaMa [25], to obtain appearance prior of occluded areas. However, the single 2D image inpainting does not consider the consistent in the real scene. Therefore, we introduce a training view selection mechanism based on regional average gradients to select the training views that are

most beneficial for the realistic inpainting of occluded areas. Specifically, when the occluded area can not be referred by any view, the occluded area is mainly modeled through the 2D inpainting results; when other views can serve for supplementary information of the occluded area in current view, the occluded area is modeled through the information from other views. We conducted detailed studies on the occluded area inpainting and concludes that, the recovered content derived from the other views is more closer to the real scene and rich in texture than the result of the individual 2D inpainter, when the supplementary information of multi-view is available.

Consequently, we propose a novel training view selection mechanism. This mechanism selects training views by evaluating the richness of texture information in the rendered results of occluded areas, aiming to optimize the inpainting results to maintain maximum consistency in the real scene. To ensure the efficiency of model training, we use the average gradient as a key indicator to measure the richness of texture information.

Notably, our view selection mechanism does not simply apply a uniform treatment strategy to the entire occluded area, but adopts a refined patching method. By dividing the occluded area into multiple sub-regions and evaluating the average gradient of these sub-regions individually, this method identifies which occluded sub-regions can learn supplementary information from other views. This refined patching method avoids misjudgment of the overall strategy and is beneficial to recover details of the occluded area realistically.

During the model training, we formulate the scene reconstruction loss with an adaptive weight adjustment based on the texture richness of occluded areas. The loss function is expressed bellow:

$$\mathcal{L} = \lambda \mathcal{L}_{NonOcc} + (1 - \lambda)\mathcal{L}_{Occ} \qquad (6)$$

$$\lambda = \begin{cases} 1 & \text{if } \nabla\hat{g} > \tau * \nabla g \\ 0 & \text{if } \nabla\hat{g} \leq \tau * \nabla g \end{cases}$$

where $\mathcal{L}_{NonOcc}$ is the reconstruction loss in non-occluded areas (Sec 3.3.1), and $\mathcal{L}_{Occ}$ is the reconstruction loss in occluded areas (Sec 3.3.2). $\lambda$ is the adaptive weight. When the average gradient $\nabla\hat{g}$ of the rendering result in a sub-region exceeds the $\tau$ times of the average gradient $\nabla g$ of the results of the 2D inpainting, $\lambda$ equals to 1; otherwise, it is 0. This formulation allows the model to adaptively adjust its inpainting strategy based on the richness of texture information.

### 3.3.1 Reconstruction Loss for Non-Occluded Areas.
The reconstruction loss function for non-occluded areas is defined as:

$$\mathcal{L}_{NonOcc} = \sum_{r \in R} (1 - \hat{P}(r))[||\hat{C}_c(r) - C(r)||_2^2 + ||\hat{C}_f(r) - C(r)||_2^2] \quad (7)$$

where $\hat{P}(r)$ is the mask value of the pixel corresponding to the ray $r$. $\hat{C}(r)$ is the predicted color of ray $r$. $C(r)$ is the actual color of the input view for ray $r$. Guided by multi-view masks, the $\mathcal{L}_{NonOcc}$ aims to constrain the scene modeling process corresponding to non-occluded areas in each input view. Due to the correlation of multi-view images, areas occluded in some views may be visible in other views. In this case, the supplementary information from

other views can be effectively exploited to reconstruct the occluded regions in the current view.

### 3.3.2 Reconstruction Loss for Occluded Areas.
The inpainting result $C_l$ by the 2D inpainter is regarded as an estimated color prior information to guide the modeling of occluded areas that lack supplementary information at all views. Specifically, the reconstruction loss for occluded areas is defined as:

$$\mathcal{L}_{Occ} = \sum_{r \in R} \hat{P}(r)[||\hat{C}_c(r) - C_l(r)||_2^2 + ||\hat{C}_f(r) - C_l(r)||_2^2] \quad (8)$$

where $C_l(r)$ is the color of the corresponding pixel in the 2D inpainting image of the ray $r$.

## 4 EXPERIMENTS AND EVALUATIONS

### 4.1 Experiment setup

#### 4.1.1 Datasets.
To evaluate our proposed 3D de-occluding method, we conduct extensive experiments on a total of 23 scenes from three datasets with different characteristics. First, we utilize the NeRF [21] LLFF dataset (8 scenes). It includes both indoor and outdoor environments, providing rich testing conditions for object segmentation and scene de-occlusion tasks. However, since the LLFF dataset does not provide benchmarks for evaluating de-occlusion tasks, we further adopt the SPIn-NeRF [22] dataset (8 scenes), which offers reference views of scenes after obstacle removal for performance evaluation. However, the scenes in the SPIn-NeRF [22] dataset are mainly captured within a small viewing angle range, and the obstacles have relatively uniform shapes and colors. To fully assess the performance of our method in datasets with different shooting angle ranges, we create a custom dataset containing 7 scenes shot within a broader viewing angle range, as shown in Figure 2. In selecting obstacles, we deliberately choose objects with various shapes, sizes, and colors. The translucent and complex-edge objects are also included. These designs aim to pose higher challenges to test our method's ability in handling scenes with a wide range of shooting angles of multi-view, and diverse obstacle characteristics.

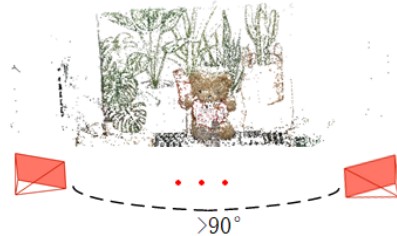

**Figure 2: Illustration for the wide range of shooting angle of the multi-view in the newly established dataset. The angle between two end views is larger than $90°$.**

#### 4.1.2 Metrics and Baselines.
To evaluate our object segmentation component and the 3D inpainting effect, we introduce evaluation metrics and propose baseline methods for comparisons:

| Input images | Ground-truth | STCN [4] | Ours |

**Figure 3: In the object segmentation task, our method is qualitatively compared with the baseline method. Compared with the STCN, the object mask obtained by our method is more precise.**

**Metrics.** To evaluate the accuracy of object segmentation in 3D scenes and the quality of the inpainting of occluded areas after obstacle removal, we employ three standard metrics: PSNR [12], LPIPS [39], and FID [10]. PSNR [12] is widely used to assess the similarity, while the higher PSNR indicates better quality. LPIPS [39] evaluates the perceptual similarity and FID [10] measures the similarity between image distributions. Lower LPIPS and FID indicate better results. These metrics are calculated by comparing the outputted rendered images with the reference images in the dataset. Particularly, due to the lack of ground-truth of the object mask for all views, we adopt an indirect way to evaluate the segmentation performance of our method. We apply our method and the segmentation baseline method to the same 3D inpainting method. After that we measure the inpainting results by metrics to assess whether our segmentation method is able to provide more precise object masks for more accurate inpaintings. During the evaluation, we average the assessment results of all views in the dataset to fully reflect the performance of test method.

**3D Object Segmentation baseline.** For the object segmentation task, we use STCN [4] as the baseline method. First, STCN [4] has been proven to have outstanding performance in video segmentation. Compared to other video segmentation methods, STCN [4] can utilize the annotation information of one frame, then quickly extend to the entire video, leading to high accurate segmentation results. Secondly, considering current works related to NeRF-based de-occlusion task, such as SPIn-NeRF [22] and NeRFIn [24], they both use STCN [4] to generate multi-view masks as part of the processing flow. Therefore, the widely application and effectiveness of STCN [4] make it suitable and reliable as the baseline method for object segmentation in the experiments.

**3D inpainting baseline.** To comprehensively evaluate the effectiveness of our method, we select a series of baseline methods related to our work for comparison. They are two image and video inpainting methods: LaMa [25] and E$^2$FGVI [17], as well as two NeRF-based 3D inpainting methods: SPIn-NeRF [22] and NeRF-In [24]. LaMa [25] is an advanced 2D image inpainting solution. It provides a meaningful reference for the 2D image inpainting. E$^2$FGVI [17] is a video inpainting method. Although it does not support the synthesis of new viewpoint images, its ability to maintain consistency between continuous frames offers an important benchmark for multi-view image inpainting results. Additionally, our method is compared with recent NeRF-based 3D inpainting

methods, SPIn-NeRF [22] and NeRF-In [24]. Both SPIn-NeRF [22] and NeRF-In [24] use depth and color images obtained from 2D inpainting methods as priors to serve the reconstruction of occluded area. Notably, NeRF-In [24] adopts color priors obtained from 2D inpainting on a single view to constrain the modeling process of occluded areas. SPIn-NeRF [22] utilizes a perceptual loss function to ensure consistency of the de-occluded areas across different views.

## 4.2 Results

### 4.2.1 Object Segmentation.

In the aspect of object segmentation, our method is compared with the baseline method STCN [4], aiming to demonstrate the advantages of our method through both quantitative and qualitative evaluations.

**Quantitative Evaluation:** We apply our segmentation method and STCN [4] to produce multi-view masks for SPIn-NeRF [22] and NeRF-In [24]. Through comparing the 3D inpainting results from the aforementioned models, we can verify which segmentation method contributes to better inpainting quality by generating more precise masks. Since the LLFF [21] dataset does not provide reference images for de-occlusion, our quantitative evaluation focuses on the SPIn-NeRF [22] dataset and our custom dataset. As shown in Table 1, the de-occlusion results guided by masks generated from our proposed segmentation component outperform those by STCN [4], across all evaluation metrics. These results demonstrate that our proposed object segmentation method achieves more precise segmentation of obstacles in multiple views.

**Qualitative Evaluation:** The qualitative evaluation is performed through intuitive visual comparisons. As shown in Figure 3, a semi-transparent bottle with flowers is segmented in the first group, while a vase is masked in the second group. The masks obtained by our object segmentation method keep more edge details, whereas the results from STCN [4] become rough. This indicates that our method has higher precision to segment the contour of obstacles. Particularly, when the occluded areas are not coherent, such as the ribbon on the semi-transparent bottle in the first group, our segmentation method can detect the ribbon more accurately, whereas STCN [4] mistakenly identifies the area between the ribbon and the bottle as part of the occluded area. This advantage comes from the depth constraint introduced in our object segmentation component. With the depth information, the network can grasp the position relationships of different sampling points in space. Since

**Table 1: Quantitative evaluation of our object segmentation method, 3D inpainting method, and baseline methods. Specifically, the fourth and sixth rows in the table show the quantitative evaluation results of the 3D inpainting tasks performed by SPIn-NeRF [22] and NeRF-In [24], guided by multi-view masks generated by STCN [4], while the third and fifth rows show the quantitative evaluation results of SPIn-NeRF and NeRF-In, guided by multi-view masks from our object segmentation method.**

| Metrics | SPIn-NeRF[22] Dataset | | | Our Dataset | | |
|---|---|---|---|---|---|---|
| | PSNR ↑ | FID ↓ | LPIPS ↓ | PSNR ↑ | FID ↓ | LPIPS ↓ |
| **No novel view synthesis** | | | | | | |
| E$^2$FGVI [17] | 29.063 | 65.477 | 0.486 | 30.662 | 51.028 | 0.125 |
| LaMa [25] | 29.053 | 58.005 | 0.479 | 30.571 | 90.223 | 0.165 |
| **NeRF based ablations** | | | | | | |
| SPIn-NeRF [22] - masks generated by our Segmentation | **29.173** | 51.366 | 0.444 | 30.573 | 99.367 | 0.157 |
| SPIn-NeRF [22] - masks generated by stcn[4] | 29.140 | 52.743 | 0.456 | 30.553 | 108.987 | 0.172 |
| NeRF-In [24] - masks generated by our Segmentation | 28.771 | 99.176 | 0.500 | 30.569 | 100.146 | 0.157 |
| NeRF-In [24] - masks generated by stcn[4] | 28.769 | 102.107 | 0.501 | 30.502 | 103.111 | 0.163 |
| Ours | 29.070 | **47.091** | **0.370** | **30.800** | **43.734** | **0.111** |

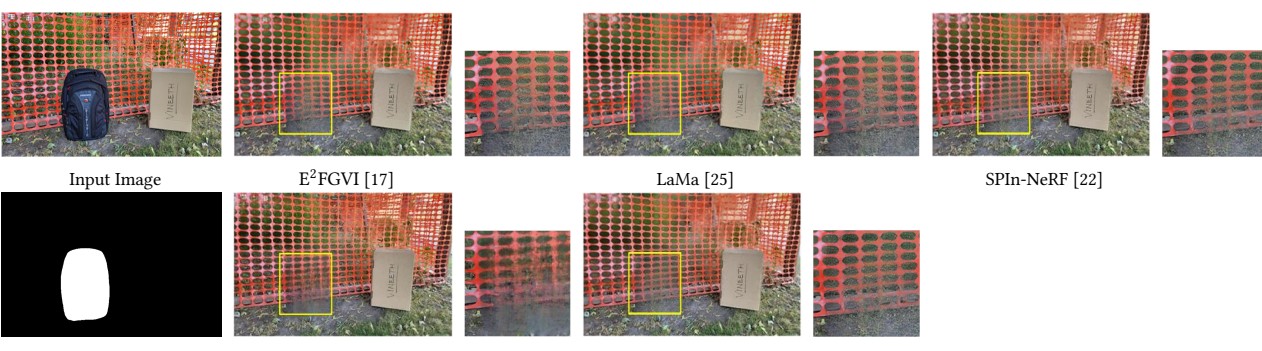

Input Image  E$^2$FGVI [17]  LaMa [25]  SPIn-NeRF [22]

Mask  NeRF-In[24]  Ours

**Figure 4: In the 3D inpainting task, we provide a qualitative comparison between our method and baseline methods in the SPIn-NeRF [22] dataset.**

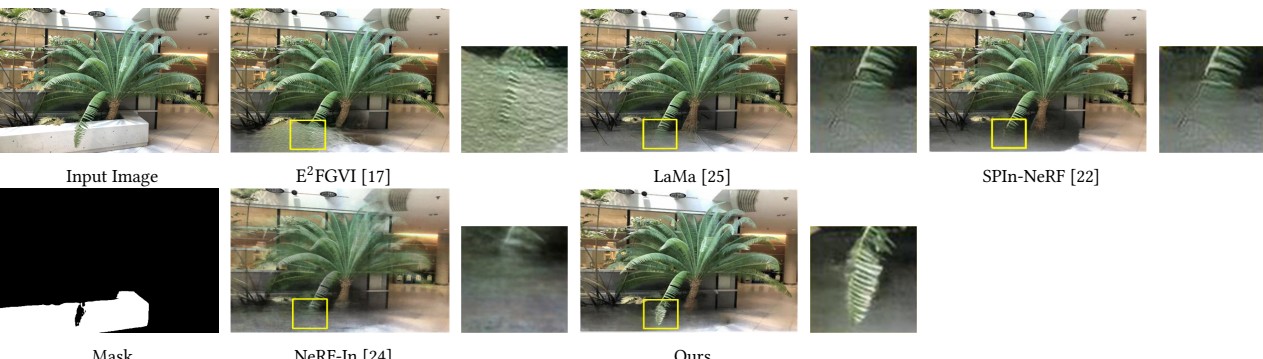

Input Image  E$^2$FGVI [17]  LaMa [25]  SPIn-NeRF [22]

Mask  NeRF-In [24]  Ours

**Figure 5: In the 3D inpainting task, we provide a qualitative comparison between our method and baseline methods in the scene of the LLFF [21] dataset.**

the sampling points on the ribbon and the points in the gap between the ribbon and the bottle have inconsistent depths, the network can distinguish these subtle depth differences during the learning process. The visual comparison presents the superiority of our object segmentation method over STCN [4] in terms of segmentation accuracy.

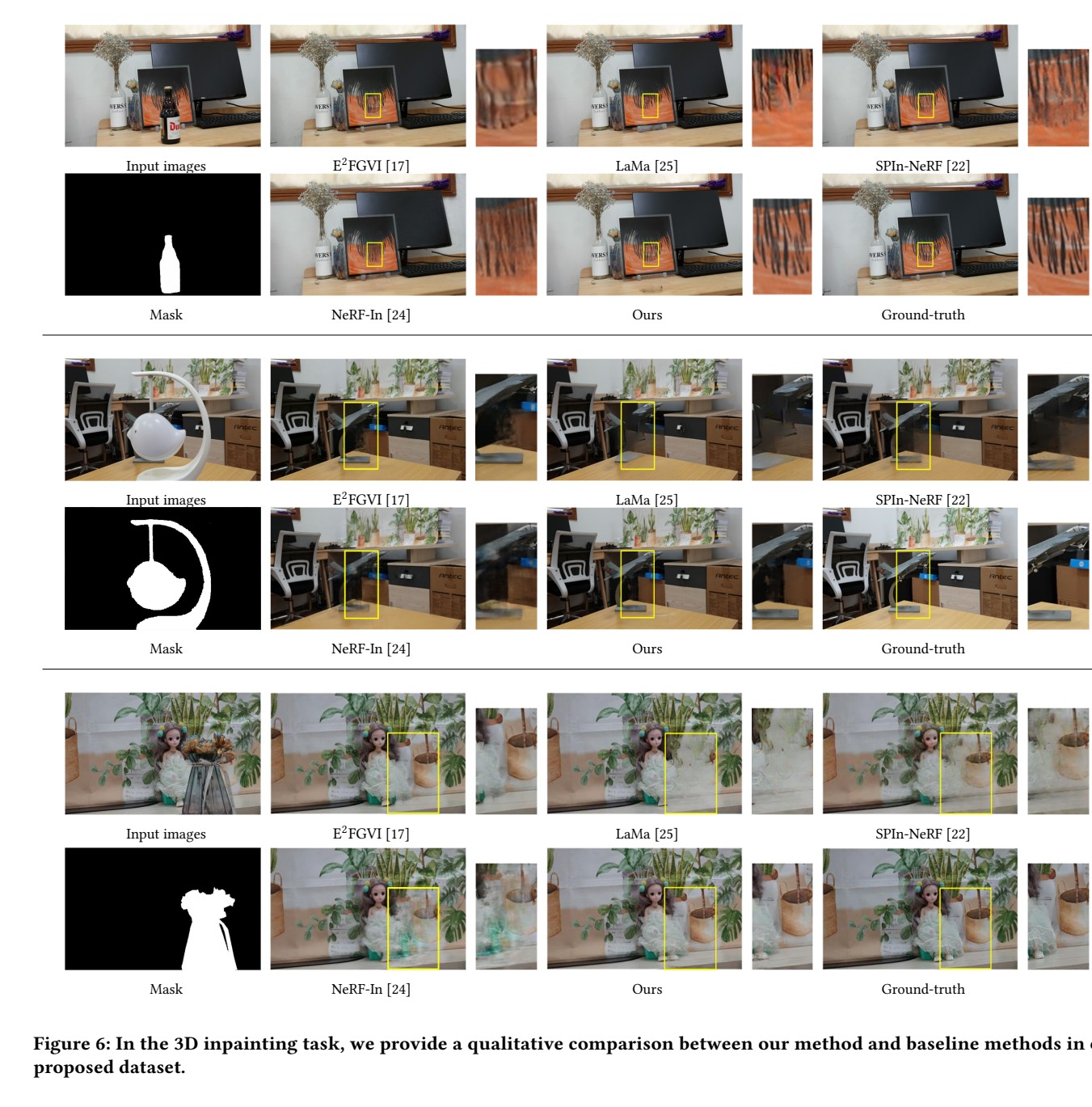

**Figure 6: In the 3D inpainting task, we provide a qualitative comparison between our method and baseline methods in our proposed dataset.**

### 4.2.2 3D Inpainting.

This section quantitatively and qualitatively evaluates our proposed 3D inpainting method against existing image and video inpainting methods (LaMa [25] and E$^2$FGVI [17]), as well as two NeRF-based 3D inpainting methods (SPIn-NeRF [22] and NeRF-In [24]) across different datasets to demonstrate the advantages of our method.

**Quantitative Evaluation:** In the SPIn-NeRF [22] dataset, we evaluate our proposed 3D inpainting method and the four recent methods across three performance metrics: PSNR [12], LPIPS [39], and FID [10]. As shown in Table 1, our 3D inpainting method

ensures reasonable results of the scene after de-occlusion, outperforming the four methods in both the LPIPS [39] and FID metrics, while is slightly inferior to SPIn-NeRF [22] in terms of the PSNR [12] for the SPIn-NeRF dataset. In our proposed dataset, which supports wider range of the multi-view, our introduced training view selection mechanism can effectively utilize information from other views to optimize the inpainting process of the occluded areas. The selection mechanism avoids adverse effects such as view inconsistency or distortion that may be caused by the 2D inpainting, while improves the fidelity of the inpainting results of occluded areas.

In the newly proposed dataset, our method outperforms all other methods in three metrics. The video inpainting method E$^2$FGVI [17], which makes use of the inter-frame correlation, also performs stably. The mask refinement strategy proposed by SPIn-NeRF [22] is not stable in different datasets, leading to a slightly lower PSNR value than E$^2$FGVI [17], but better than LaMa [25] in our dataset.

**Qualitative Evaluation:** In Figure 4, we present the qualitative results of our 3D inpainting method for the SPIn-NeRF [22] dataset, demonstrating its ability to remove the obstacle in scenes captured in the narrow range of multi-view. Additionally, in Figure 5, we show the results in the LLFF dataset. Compared with other methods, our method effectively preserves the details outside the mask. The leaf area of the tree has been kept in our result. However, SPIn-NeRF [22] and NeRF-In [24] involve blurring to the leaf area. In Figure 6, we display the qualitative results in our dataset. The three groups of results in Figure 6 successively show that our method can reasonably model regions and sufficiently employ supplementary information across views for inpainting of the occluded area. Our results are closer to the ground-truth. For the second group where a white desk lamp is regarded as the obstacle, our method can still achieve realistic inpainting result for the largely occupied scene. For the third group where a semi-transparent bottle is masked, our method can reappear the body of the toy, whereas other solutions involve much blurring distortion and cannot reveal the details behind the obstacle.

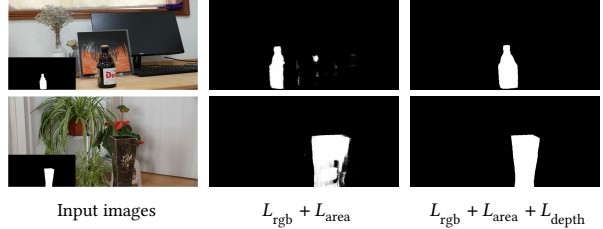



Input images      $L_{\text{rgb}} + L_{\text{area}}$      $L_{\text{rgb}} + L_{\text{area}} + L_{\text{depth}}$



**Figure 7: Ablation Studies to illustrate the importance of the depth loss function for the segmentation model**

### 4.3 Ablation Studies

Here, we verify the performance discussed in Sec 3.2 that introducing depth information can significantly improve multi-view segmentation results. We compare the segmentation results of our method with and without adding depth priors. As shown in Figure 7, the color loss and the area encoding loss can not guarantee the accuracy of the predicted multi-view masks by a single user annotation mask. With the depth loss, the scene geometry and appearance can be correctly modeled for more accurate segmentation result.

## 5 CONCLUSION

In this paper, we propose a novel de-occlusion method for 3D scenes based on the Neural Radiance Fields (NeRFs). We introduce an object segmentation component to generate the obstacle masks for multiple views from a given mask in one view, which significantly reduces the labeling effort. The following 3D inpainting component equipped with a view selection mechanism sensitive to the average

regional gradient has improved the appearance consistency of the inpainting area, enabling users to seamlessly remove obstacles in NeRF-modeled scenes. Our experiments validate the advantages of our method in terms of segmentation accuracy and fidelity of occluded area inpainting results. Furthermore, we introduce a real-world dataset captured from a wider range of viewpoints, providing strong support for in-depth research into the problem of 3D scene de-occlusion.

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
