# OpenReview forum: "3D Scene De-occlusion in Neural Radiance Fields: A Framework for Obstacle Removal and Realistic Inpainting"
_acmmm.org/ACMMM/2024/Conference — MM2024 Poster_

### Official Review · Reviewer_eVps · 2024-05-24

**Rating:** 6
**Confidence:** 3

**Summary:**

This paper uses Neural Radiance Fields (NeRFs) to realize 3D de-occlusion and inpainting for rendered scene images.

**Strengths:**

This paper
1) proposes to use depth priors as scene modeling constraints to improve the multi-view object segmentation. With adding depth information, the qualities of segmentation results are improved;
2) proposes to generate the obstacle masks for multiple views from a user-made mask in one view to significantly reduce the labeling effort;
3) establishes a new dataset containing several scenes shots within a broader viewing angle range, which can provide strong supports for in-depth research into the problem of 3D scene de-occlusion.

**Limitations:**

The de-occlusion and inpainting results are impressive. However, in all tested datasets, there is only one occluded area on the image. The authors are expected to show de-occlusion results with multiple occluded areas (like the situation in Fig. 4). Can a user mark more obstacle objects simultaneously on one view to obtain the required mask in the proposed framework?

**Suitability:**

3

---

### Official Review · Reviewer_bsbW · 2024-05-24

**Rating:** 3
**Confidence:** 2

**Summary:**

The authors propose a framework for the De-occlusion in NeRFs by leveraging a view single mask. The framework consists of e a new region encoding network and a training view selection mechanism in inpainting of occluded area. And the authors a new 3D de-occlusion dataset containing scene images both with and without the obstacle for further researhes.

**Strengths:**

1. This proposed method achieves good performances in inpainting tasks, which is reflected in both the quantitive and visualization results.
2. The paper is well written and easy to understand.

**Limitations:**

1. The effects of the proposed method rely highly on the LaMa model, making the contributions somewhat limited.

**Suitability:**

2

---

### Official Review · Reviewer_J3nu · 2024-05-24

**Rating:** 4
**Confidence:** 3

**Summary:**

This article introduces an innovative Neural Radiance Fields (NeRF) framework for 3D inpainting and presents a specific 3D dataset tailored for this application, with a particular emphasis on mask selection.

**Strengths:**

The primary strengths of this article lie in its proposal of an inpainting algorithm that requires only one masked image. The authors conduct evaluations and comparisons with recent state-of-the-art methods, while also extending the current dataset to encompass a wide range of use cases. The article is well-written and easy to understand

**Limitations:**

Despite these contributions, the article does not compare its results with the work presented in "Removing Objects From Neural Radiance Fields," nor does it utilize the dataset provided in that study. Although the article highlights the advancement of automatic mask selection in multiple views, the aforementioned study introduces a mask refinement process that refines a raw bounding box to the precise edges of the object. Overall, it would be valuable to understand if there are significant drawbacks to employing the proposed method with only one mask compared to the state-of-the-art approach with classical masking strategy.

**Suitability:**

3

---

### Official Review · Reviewer_Mt6B · 2024-05-25

**Rating:** 3
**Confidence:** 3

**Summary:**

This paper proposes a NeRF framework for obstacle removal and high-quality reconstruction of occluded regions in wide-range multiviews.

**Strengths:**

The paper is well structured and the experimental results are superior to the existing SOTA.

**Limitations:**

1) As I understood it, the paper proposes consistent obstacle removal and realistic inpainting in images from various viewpoints, but the experimental results are mostly presented only for one image. It seems difficult to judge whether the removal and inpainting are consistent.

2) Regarding the quantitative comparison results in Section 4.2.1, I think it would be helpful to clarify that it is for a part of Table 1, not all of it, and it would be better to separate the table if there is no problem with the volume.
However, the experiment is an indirect comparison of in-painting performance, not a direct experiment of object segmentation, but it seems that there is ground truth in Figure 3, so I think a direct comparison is possible.

3) Following up on comment 1), is the result in Figure 3 a comparison of one of the multiple 2D masks shown in the Object segmentation component part of Figure 1? It would be important to know if the detail of the ribbon remains the same across the multiple results, as the authors describe.

4) From the description of the experiment, I understood that the SPIn-NeRF dataset has GT, but no ground truth is presented in Figure 4. Is there a reason for this?

5) Regarding the experimental results in Figure 5, I wonder if the comparative papers in [17], [25], [22], and [24] used the same mask as in Figure 5. If the mask was formed as in the figure, I think the foliation should be similar.

**Suitability:**

3

---

### Meta-Review · Area_Chair_T5mK · 2024-07-01

**Recommendation:** Accept (Poster)
**Confidence:** 4

**Metareview:**

Overall, authors have clearly stated their motivation and proposal, allowing the reviewers to understand the paper better. Furthermore, the authors have explained their methodology and presented their results well.

In terms of responding to reviewers' comments, the authors have provided sufficient explanation to debunk the concerns stated by reviewers and have conducted extra experiments for validating the performance of their method and clarify uncertainties questioned by the reviewers.